# Skin Lesion Classification on Imbalanced Data Using Deep Learning with Soft Attention

**DOI:** 10.3390/s22197530

**Published:** 2022-10-04

**Authors:** Viet Dung Nguyen, Ngoc Dung Bui, Hoang Khoi Do

**Affiliations:** 1School of Electrical and Electronic Engineering, Hanoi University of Science and Technology, Dai Co Viet, Ha Noi 100000, Vietnam; 2Faculty of Information Technology, University of Transport and Communications, Ha Noi 100000, Vietnam

**Keywords:** skin lesions, classification, deep learning, soft-attention, imbalance

## Abstract

Today, the rapid development of industrial zones leads to an increased incidence of skin diseases because of polluted air. According to a report by the American Cancer Society, it is estimated that in 2022 there will be about 100,000 people suffering from skin cancer and more than 7600 of these people will not survive. In the context that doctors at provincial hospitals and health facilities are overloaded, doctors at lower levels lack experience, and having a tool to support doctors in the process of diagnosing skin diseases quickly and accurately is essential. Along with the strong development of artificial intelligence technologies, many solutions to support the diagnosis of skin diseases have been researched and developed. In this paper, a combination of one Deep Learning model (DenseNet, InceptionNet, ResNet, etc) with Soft-Attention, which unsupervisedly extract a heat map of main skin lesions. Furthermore, personal information including age and gender are also used. It is worth noting that a new loss function that takes into account the data imbalance is also proposed. Experimental results on data set HAM10000 show that using InceptionResNetV2 with Soft-Attention and the new loss function gives 90 percent accuracy, mean of precision, F1-score, recall, and AUC of 0.81, 0.81, 0.82, and 0.99, respectively. Besides, using MobileNetV3Large combined with Soft-Attention and the new loss function, even though the number of parameters is 11 times less and the number of hidden layers is 4 times less, it achieves an accuracy of 0.86 and 30 times faster diagnosis than InceptionResNetV2.

## 1. Introduction

### 1.1. Problem Statement

Skin cancer is one of the most common cancers leading to worldwide death. Every day, more than 9500 [1] people in the United States are diagnosed with skin cancer. Otherwise, 3.6 [1] million people are diagnosed with basal cell skin cancer each year. According to the Skin Cancer Foundation, the global incidence of skin cancer continues to increase [2]. In 2019, it is estimated that 192,310 cases of melanoma will be diagnosed in the United States [2]. On the other hand, if patients are early diagnosed, the survival rate is correlated with 99 percent. However, once the disease progresses beyond the skin, survival is poor [2]. Moreover, with the increasing incidence of skin cancers, low awareness among a growing population, and a lack of adequate clinical expertise and services, there is a need for effective solutions.

Recently, deep learning particularly, and machine learning in general algorithms have emerged to achieve excellent performance on various tasks, especially in skin disease diagnosis tasks. AI-enabled computer-aided diagnostics (CAD) [3] has solutions in three main categories: Diagnosis, Prognosis, and Medical Treatment. Medical imaging, including ultrasound, computed tomography, magnetic resonance imaging, and X-ray image is used extensively in clinical practice. In Diagnosis, Artificial Intelligence (AI) algorithms are applied for disease detection to save progress execution before these diagnosis results are considered by a doctor. In Prognosis, AI algorithms are used to predict the survival rate of a patient based on his/her history and medical data. In Medical Treatment, AI models are applied to build solutions to a specific disease; medicine revolution is an example. In various studies, AI algorithms have provided various end-to-end solutions to the detection of abnormalities such as breast cancer, brain tumors, lung cancer, esophageal cancer, skin lesions, and foot ulcers across multiple image modalities of medical imaging [2].

To adapt the rise in skin cancer cases, AI algorithms over the last decade has a great performance. Some typical models that can be mentioned are DenseNet [4], EfficientNet [5], Inception [6,7], MobileNets [5,8,9], Xception [10], ResNet [11,12], and NasNet [13]. Some of these models which have been used as a backbone model in this paper will be discussed in the Related Work section.

### 1.2. Related Works

Skin lesion classification is not a new area, since there are many great performance models constructed, recent years. The skin classification approaches can be divided into two main approaches: Deep Learning and Machine Learning (as shown in Table 1). Both approaches gain great performance. Data Augmentation and Feature Extractor, otherwise are two main supporters that make the model better.

#### 1.2.1. Deep Learning Approach

In Deep Learning, one of the most cutting-edge technologies used is Soft-Attention, as stated in [1]. Soumyyak et al. constructed several models formed by a combination of a backbone model including DenseNet201 [4], InceptionResNetV2 [6], ResNet50 [11,12], VGG16 [27] and Soft-Attention layer. Their approach adds the Soft-Attention layer at the end or the middle of the backbone model. For ResNet50 and VGG16, the Soft-Attention layer is added after the third residual block and CNN block, respectively. DenseNet201 and InceptionResNetV2 then concatenate with Soft-Attention before a fully-connected layer and then soft-max layer. Soumyyak et al.’s proposed method gained great performances and also outperformed many other studies with an accuracy of 0.93 and a precision of 0.92. However, using data augmentation on an imbalanced dataset resulted in subpar classification classify with respect to the classes; therefore, their model obtained a recall and F1-score of 0.71 and 0.75, respectively. In this research, our proposed method also considers this problem and solves it.

Using the above-mentioned backbones has been attempted previously. Rishu Garg et al. [14] used a transfer learning approach with a CNN-based model: ResNet50 and VGG16 which are pretrained with an ImageNet data set. In addition, they also use data augmentation to avoid an imbalance occurring in the data set. Histogram equalization is also used to increase the contrast of skin lesions before being fed into machine learning algorithms including Random Forest, XGBoost, and Support Vector Machine. Histogram equalization can be considered as a heat map that takes the main feature as the number of occurrences of the same value pixel. This approach also gain great performances with an accuracy of 0.90 and precision of 0.88. However, this approach can be biased since only one skin image of the dataset contains the skin lesion at the center and the background skin, and the histogram may treat the background with increased numbers of occurrence with respect to the same pixel value. In this research study, our proposed method used Soft-Attention, which can create a heat map feature of the lesion. Otherwise, Rishu Garg et al.’s proposed method also faced the problem of imbalanced classification due to an imbalanced dataset with the F1-score and recall of 0.77 and 0.74, respectively.

Instead of using the entire imbalanced data set, Abayomi-alli et al. decided to separate the dataset into two subsets: one contains only melanoma and the other one contains the rest [24]. Before feeding the data to classify melanoma, training data are then augmented by the SMOTE method. SMOTE creates artificial instances by oversampling the minority class. SMOTE recognizes k-minority class neighbors that are near each minority class sample by using the covariance matrix. This approach obtained an accuracy, recall, and F1-score of 0.92, 0.87, and 0.82, respectively.

Amirreaza et al. [15] did not only use the backbone model mentioned above but also used the InceptionV3 [6] model. In this research study, datasets HAM10000 and *PH*^2^ are combined to create an eight-class dataset. Before being fed into Deep CNN models, the image was resized to (224, 224) for DenseNet201, ResNet152, InceptionResNetV2, and (229, 229) for InceptionV3. The best AUC values for melanoma and basal cell carcinoma are 0.94 (ResNet152) and 0.93 (DenseNet201).

Another paper that uses backbone models is [16], in which Hemanth et al. decided to use EfficientNet [28] and SeNET [29] instead and the CutOut [30] method, which involves creating holes of different sizes on images, i.e., technically making a random portion of image inactive during the data augmentation process. Although this approach obtained an accuracy of 0.88, it may be biased due to the CutOut method since this method can create a hole overlap in the skin lesion field. The method’s accuracy is also low due to the data-augmentation process.

Otherwise, Ref. [17] also used a Deep Convolution Neural Network, and Peng Yao et al. used RandArgument, which crops an image into several images from a fixed size; DropBlock, which is used for regularization, Multi-Weighted New Loss, which is used for dealing with the imbalanced data problem; end-to-end Cumulative Learning Strategy, which can more effectively balance representation learning; and classifier learning, without additional computational costs. This approach obtained an accuracy of 0.86. Although this approach figureed out the data imbalance problem, the result of obtaining a low accuracy may due to RandArgument. If the skin lesion part of the image is quite big or small, the cropped image may only contain skin or the lesion is spread out in the entire image.

Another state-of-the-art method is GradCam and Kernel SHAP [18]. Kyle Young et al. created an agnostic model, which includes local interpretable methods that can highlight pixels that the trained network deems relevant for the final classification. In that research study, they used three datasets containing HAM10000, BCN-20000, and MSK. Before feeding into the models, images are preprocessed by binarization with a very low threshold to find the center of mass. This approach obtained an AUC of 0.85.

On the other hand, there are also many state-of-the-art methods with great performance on skin lesion classification. The Student-and-Teacher Model is also a high-performance model introduced in 2021 [19], and it is created by Xiaohan Xing et al. as a combination of two models that share memories with the other model. Therefore, the models can take full advantage of what others learn. The Student-and-Teacher model obtained an accuracy of 0.85; however, the precision and F1-score are quite low, resulting in a value of 0.76.

SkinLinkNet [20] and WonderM [21] are both tested the effect of segmentation on skin lesion classification problems created by Amirreza et al. and Yeong Chan et al., respectively. In WonderM, the method used is to pad the image so that the image has an increase in shape from (450, 600) to (600, 600). In SkinLinkNet, the image is instead resized down to (448, 448). Both SkinLinkNet and WonderM used UNet to perform the segmentation task, although they used EfficientNetB0 and DenseNet to perform the classification task. This approach obtained an AUC of 0.92.

Another approach is to use metadata, including gender, age, and capturing positions, as stated in [22] by Nil Gessert et al. Metadata are fed into a fully connected neural network after encoded into a one-hot vector. All missing data points with respect to age are set to 0. To overcome the missing data problem, the research study applied one-hot encoding to the group, but the initial validation resulted in poor performance then when numerical encoding was applied. The metadata are then fed into two block networks, each one containing a Dense Layer, Batch Normalization, am ReLU activation function, and a Dropout. After all the feature vectors were extracted, the image was then concatenated with the feature vector extracted from metadata. Otherwise, data augmentation was also applied. This approach obtained a recall of 0.74. The low recall may be due to the imbalanced data set.

Abnormal, skin lesion segmentation, on the other hand, also plays an important role in skin lesion classification. Nawaz et al. created a framework for Melanoma segmentation [25]. Their proposed method is a Unet model but used DenseNet77 as the backbone, and all residual blocks were changed into dense block, which contains a sequence of Convolution and Average Pooling. This melanoma segmentation approach obtain an accuracy of 0.99. Kadry et al. used a Unet model with a VGG16 deep convolution layer by pooling on the skip connection. This approach can completely extract the entire lesion, although there was an overlap observed with hair. This approach obtained an accuracy of 0.97.

#### 1.2.2. Machine Learning Approach

In Machine Learning, there are also many approaches. Since the image’s data are quite complex for machine learning algorithms, using feature extractors or feature preprocessing for transformation to another form of data is recommended.

Random Forest, XGBoost, and Support Vector Machines are tested by [14] of Rishu Garg et al. In this approach, the data are fed directly into the Machine Learning algorithm and shows no promising results; therefore, Rishu Garg et al. did not show the results of the used machine learning algorithm.

In addition, Deep Isolation Forest is applied before the soft-max activation of the deep learning model to detect the distribution of skin lesion images, as stated in [31] by Amirreza Rezvantalab et al. In the Deep Isolation Forest, an feature extractor is applied by using CNN to learn the main pattern of the image. After that, the feature map is then fed into K isolation forest estimators by using bagging algorithms. The Deep Isolation Forest obtained an accuracy of 0.9 and a confidence of 0.86. However, the AUC is only 0.74, and this may due to the limitation of the machine learning algorithm.

Matrix transformation is also applied before the soft-max activation function in [23] by Michele Alberti et al. In this approach, the image is fed into a general model by using a sequence of residual block. The feature maps created from those above the residual block is then fed into Global Average Pooling to create a feature vector. This feature vector is then extracted by CNN-1D and transformed by Discrete Fourier Transformation (DFT) as a filter before proceeding to the soft-max layer.

### 1.3. Proposed Method

In this research, a new model is constructed from the combination of:-Backbone model including DenseNet201, InceptionResNetV2, ResNet50/152, NasNetLarge, NasNetMobile, and MobileNetV2/V3;-Using metadata including age, gender, localization as another input of the model;-Using Soft-Attention as a feature extractor of the model;-A new weight loss function.

## 2. Materials and Methods

### 2.1. Materials

#### 2.1.1. Image Data

The data set used in this paper is the HAM10000 data set published by the Havard University Dataverse [32]. There are a total of 7 classes in this data set containing Actinic keratoses and intraepithelial carcinoma or Bowen’s disease (AKIEC), basal cell carcinoma (BCC), benign keratosis-like lesions (solar lentigines / seborrheic keratoses andchen-planus like keratoses, BKL), dermatofibroma (DF), melanoma (MEL), melanocytic nevi (NV), and vascular lesions (angiomas, angiokeratomas, pyogenic granulomas and hemorrhage, VASC). The distribution of the data set is shown in Table 2 below:

More than 50% of lesions are confirmed through histopathology (HISTO); the ground truth for the rest of the cases is either follow-up examination (FOLLOWUP), expert consensus (CONSENSUS), or confirmation by in vivo confocal microscopy (CONFOCAL). On the other hand, before being used for training, the whole data are shuffled and then split into two parts. Here, 90% and 10% of the data is used for training and validating respectively. Images in this data set have the type of *RGB* and shape of (450, 600). However, each backbone needs the different input sizes of images as well as the range of pixel value (as shown in Figure 1).

#### 2.1.2. Metadata

The HAM10000 data set [32] also contains the metadata of each patient including gender, age, and the capturing position, as illustrated in Table 3.

### 2.2. Methodology

#### 2.2.1. Overall Architecture

The whole architecture of the model is represented in the Figure 2. The model takes two inputs including Image data and Metadata. The metadata branch otherwise is preprocessed before feeding into a dense layer; then, it concatenates with the output of the Soft-Attention layer.

Figure 3 illustrates the overall structures of the combination of backbone models and Soft-Attention, which is used in this research. In detail, the combination of DenseNet201 and Soft-Attention is formed by replacing the three last (DenseBlock, Global Average Pooling, and the fully connected layer) with the Soft-Attention Module. Similarly, ResNet50 and ResNet152 also replaced the last three (Residual Block, Global Average Pooling, and the fully connected layer) with the Soft-Attention module. InceptionResNetV2, on the other hand, replaces the average pool and the last dropout with the Soft-Attention Module. The last two, Normal Cell in NasNetLarge is replaced with the Soft-Attention module.

Figure 4, on the other hand, shows the detailed structure of the mobile-based mobile and its combination with Soft-Attention. All of the MobileNet versions combine with the Soft-Attention module by replacing the two last convolution layers 1 × 1 with the Soft-Attention module. The NasNetMobile, otherwise, combines with the Soft-Attention module by replacing the last normal cell.

#### 2.2.2. Input Schema

Image preprocessing is an essential part of the training process because of its ability to extract the main pattern of an image. In this stage, the image can be changed to the other color channel so that the main feature is separated from the useless part. Image Retrieval has significantly created a vector that represents the main feature of an image. These image retrieval techniques can include energy compaction, primitive pattern units, etc. Shervan Fekri-Ershad et al. created a feature vector by calculating the element-wise product of the histogram vector in each channel of an image [33]. Then, by comparing the Euclidean distance between this feature vector and the average feature vector of the entire dataset with a thresh hold, they can extract the skin portion of the image.

In this research, the image data are both augmented for all classes, the number of images increases to 18,015 images , and it keeps the original form. Before feeding into the backbone model, the images are pre-processed by the input requirement of each model. DenseNet201 [4] requires the input pixels values to be scaled between 0 and 1 and each channel is normalized with respect to the ImageNet data set. In Resnet50 and Resnet152 [11,12], the images are converted from *RGB* to *BGR*; then, each color channel is zero-centered with respect to the ImageNet data set, without scaling. InceptionResNetV2 [28], on the other hand, will scale input pixels between −1 and 1. Similarly, three versions of MobileNet [5,8,9], NasNetMobile and NasNetLarge [13] require the input pixel is in range of −1 and 1.

On the other hand, the metadata are also used as another input. In the research [22], they decide to keep the missing value and set its value to 0. The sex and anatomical site are categorically encoded. The age, on the other hand, is numerically normalized. After processing, the metadata are fed into a two-layer neural network with 256 neurons each. Each layer contains batch normalization, a ReLU [34] activation, and dropout with *p* = 0.4. The network’s output is concatenated with the CNN’s feature vector after global average pooling. Especially, they use a simple data augmentation strategy to address the problem of missing values in metadata. During training, they randomly encode each property as missing with a probability of *p* = 0.1.

In this research, the unknowns are kept as a type as discussed in the Metadata section. Sex, anatomical site, and age are also category encoded and numerically normalized, respectively. After processing, the metadata are then concatenated and fed into a dense layer of 4096 neurons. Finally, this dense layer is then concatenated with the output of Soft-Attention which is then discussed in the Soft-Attention section. The Input schema is described in Figure 5.

#### 2.2.3. Backbone Model

In this paper, the backbone models used in this paper are DenseNet201 [4], Inception [6], MobileNets [5,8,9], ResNet [11,12], and NasNet [13]. The combination of DenseNet201, InceptionResNetV2, and the Soft-Attention layer are both tested by the previous paper [1] with a great performance. Otherwise, Resnet50 also well classifies but with much fewer number of parameters and less depth than based on its F1-score and precision stated. Therefore, in this paper, the performance of the model Resnet152 and NasnetLarge models, which have more parameters and depth, is analyzed. On the other hand, three versions of MobileNet and the NasnetMobile will also be analyzed, which has fewer parameters (as shown in Table 4) and depth.

#### 2.2.4. Soft-Attention Module

Soft-Attention has been used in various applications: image caption generation such as [35] or handwriting verification [36]. Soft-Attention can ignore irrelevant areas of the image by multiplying the corresponding feature maps with low weights. Soft-Attention is described in Equation (Equation 1).
(1)$$f_{sa}=\gamma t\sum_{k=1}^{K}softmax\left(W_{k}*t\right)$$

Figure 6 shows the two main steps of applying Soft-Attention. Firstly, the input tensor is put in grid-based feature extraction from the high-resolution image, where each grid cell is analyzed in the whole slide to generate a feature map [37]. This feature map called $t\in R^{h\times w\times d}$ where h,w,andd is the shape of tensor generated by a Convolution Neural Network (CNN), is then input to a 3D convolution layer whose weights are $W_{k}\in R^{h\times w\times d\times K}$. The output of this convolution is normalized using the soft-max function to generate *K* (a constant value) attention maps. These *K* attention maps are aggregated to produce a weight function called α. This α function is then multiplied with feature tensor *t* and scaled by γ, which is a learnable scalar. Finally, the output of the Soft-Attention function $f_{sa}$ is the concatenation of the beginning feature tensor *t* and the scaled attention maps.

In this research, the Soft-Attention layer is applied in the same way in [1]. The Soft-Attention module is described in Figure 7.

After feeding into the ReLU function layer, the heat feature map is processed in two paths. The first path is the two-dimensional Max Pooling. In the second path, the feature map, on the other hand, is fed into the Soft-Attention layer before the two-dimensional Max Pooling. After all, these two paths are then concatenated and fed into a ReLU layer with a dropout with the probability of 0.5.

#### 2.2.5. Loss Function

The loss function used in this paper is categorical cross-entropy [38]. Consider $X=[x_{1},x_{2},\ldots ,x_{n}]$ as the input feature, $\theta =[\theta_{1},\theta_{2},\ldots ,\theta_{n}]$. Let *N*, and *C* be the number of training examples and number of classes respectively. The categorical cross-entropy loss is presented in Equation (Equation 2):(2)$$L\left(\theta ,x_{n}\right)=-\frac{1}{N}\sum_{c=1}^{C}\sum_{n=1}^{N}W_{c}\times y_{n}^{c}\times \log\left({\hat{y}}_{n}^{c}\right)$$
where ${\hat{y}}_{i}^{c}$ is the output of the model and $y_{i}^{c}$ is the target that the model should return, and $W_{c}$ is the weight of class *c*. Since the data sets face the imbalanced problem, then class weight for the loss is applied. In this research, both the original weight and a new weight formula are implemented. Originally, the weight is calculated by taking the inverse of the percentage that each class accounts for. The new weight formula is described in the Equations (Equation 3) and (Equation 4). This weight formula is the original weight multiplied by the inverse of the number of classes in the data set which makes the training more balanced. It is inspired by the “balanced” heuristic proposed by Gary King et al. [39].
(3)W=N⊙D
(4)$$D=\left[\begin{array}{cccc} \frac{1}{C\times N_{1}} & \frac{1}{C\times N_{2}} & \ldots & \frac{1}{C\times N_{n}} \end{array}\right]=\frac{1}{C}⊙\left[\begin{array}{cccc} \frac{1}{N_{1}} & \frac{1}{N_{2}} & \ldots & \frac{1}{N_{n}} \end{array}\right]$$
where *N* is the number of the training samples, *C* is the number of classes, and $N_{i}$ is the number of samples in each class *i*. *D* is the matrix that contains the inverse of $C\times N_{i}$.

## 3. Results

### 3.1. Experimental Setup

#### 3.1.1. Training

Before training, the data set is split into two subsets for training (90%) and validation (10%). The test set, otherwise is provided by the HAM10000 data set, and it contains 857 images. To analyze the effect of augmented data on the model, before the training; the image data are augmented to 53,573 images by the following technique:-Rotation range: rotate the image in an angle range of 180.-Width and height shift range: Shift the image horizontally and vertically in a range of 0.1, respectively.-Zoom range: Zoom in or zoom out the image in a range of 0.1 to create new image.-Horizontal and vertical flipping: Flipping the image horizontally and vertically to create a new image.

Otherwise, all of the models are trained with the Adam optimizer [40] with the learning rate of 0.001 which is reduced by a factor of 0.2 to a minimum learning rate of $0.1\times 10^{6}$, and the epsilon is set to 0.1. The initial epochs are set to 250 epochs, and the Early Stopping is also applied to stop the training as the accuracy of the validation set does not increase after 25 epochs. The batch size is set to 32.

#### 3.1.2. Tools

TensorFlow and Keras are two of the most popular frameworks to build a deep learning models. In this research, Keras based on TensorFlow is used to build, and clone the backbone model which is pre-trained with the Image-Net data set. Otherwise, the models are trained by NVIDIA RTX TitanV, and the data set is pre-processed with the CPU Intel I5 32 processors, and RAM 32 GB. In detail, the GPU is set up with CUDA 11.6, cuDNN 8.3, and ChipSRT as the requirement of TensorFlow version 2.7.0.

#### 3.1.3. Evaluation Metrics

The model is evaluated by using the confusion matrix and related metrics. Figure 8 illustrates the presentation of a 2×2 confusion matrix used for class 2. Consider a confusion matrix *A* with *C* number of classes. Let $A^{i}$ and $A^{j}$ be the set of *A* rows and columns respectively, therefore $A_{k}^{i}$ is the element at row *i* and column *k*
$$A=\left[\begin{array}{cccc} a_{11} & a_{12} & \ldots & a_{1j}\\ a_{21} & a_{22} & \ldots & a_{2j}\\ \vdots & \vdots & & \vdots \\ a_{i1} & a_{i2} & \ldots & a_{ij} \end{array}\right]$$

The True Positive (*TP*) of all classes in this case is the main diagonal of the matrix *A*. The following methods are used to calculate the False Positives (*FP*), False Negatives (*FN*), and True Negatives (*TN*) of all classes:(5)$$FP=-TP+\sum_{k=1}^{i}A_{k}^{i}$$
(6)$$FN=-TP+\sum_{k=1}^{j}A_{k}^{j}$$
(7)$$TN_{c}=\sum_{i=1}^{C}\sum_{j=1}^{C}a_{ij}-\left[\sum_{k=1}^{i}A_{i=ck}^{i}+\sum_{k=1}^{j}A_{j=ck}^{j}\right]+a_{i=cj=c}\Rightarrow TN=\left[\begin{array}{cccc} TN_{1} & TN_{2} & \ldots & TN_{c} \end{array}\right]$$

Then, the model is evaluated by the following metrics:(8)$$\mathrm{Sensitivity}\;\left(\mathrm{Sens}\right)=\frac{TP}{TP+FN}$$
(9)$$\mathrm{Specificity}\;\left(\mathrm{Spec}\right)=\frac{TN}{TN+FP}$$

Sensitivity (Equation (Equation 8)) and specificity (Equation (Equation 9)) mathematically describe the accuracy of a test that identifies a condition’s presence or absence. Sensitivity, also known as the true positive rate, is the likelihood that a test will result in a true positive, whereas specificity, also known as the true negative rate, is the likelihood that a test will result in a true negative.
(10)$$\mathrm{Precision}=\frac{TP}{TP+FP}$$
(11)$$\mathrm{F1}-\mathrm{score}=\frac{2\times TP}{2\times TP+FP+FN+TN}\;$$

Precision (Equation (Equation 10)) or positive predictive value (PPV) is the probability of a positive test conditioned on both truly being positive or negative. F1-score (Equation (Equation 11)), on the other hand, refers to the harmonic mean of precision and recall, which means the higher the F1-score is, the higher both precision and recall are. Besides, the expected value of precision, F1-score, and recall are also applied because of the multi-class problem.
(12)$$\mathrm{Accuracy}=\frac{TP+TN}{TP+FP+FN+TN}$$
(13)$$\mathrm{Balanced}\;\mathrm{Accuracy}=\frac{\mathrm{Sens}+\mathrm{Spec}}{2}$$

The last metric is the *AUC* (as shown in Figure 9) score standing for Area Under the Curve which is the Receiver Operating Curve (ROC) that indicates the probability of *TP* versus the probability of *FP*.

### 3.2. Discussion

According to Table 5, it is clear that the model trained with metadata has a higher accuracy than the model trained with augmented data only. While InceptionResNetV2 and DenseNet201 trained with augmented data have an accuracy of 0.79 and 0.84, respectively, their training with metadata are 0.90 and 0.89, respectively. Furthermore, Resnet50 trained with metadata data has the accuracy that outperforms the Resnet50 trained with augmented data and is twice as high as ResNet152 trained with metadata. On the other hand, mobile models including MobileNetV2, MobileNetV3Large, and NasNetMobile, even though they have a much smaller number of parameters and depth than the other model, they have quite good accuracy scores of 0.81, 0.86 and 0.86, respectively.

**Table 5 sensors-22-07530-t005:** Accuracy of all models. ACC stands for accuracy. AD stands for augmented data; this indicates that the model is trained with augmented data. MD stands for metadata, which indicates that the model is trained with metadata. The bold numbers highlight the highest performance. These results are calculated from the confusion matrix, the two highest model confusion matrices are DenseNet201 and InceptionResNetV2 (as shown in Figure 10 and Figure 11).

Model	ACC (AD)	ACC (MD)
InceptionResNetV2	0.79	**0.90**
DenseNet201	0.84	0.89
ResNet50	0.76	0.70
ResNet152	0.81	0.57
NasNetLarge	0.56	0.84
MobileNetV2	0.83	0.81
MobileNetV3Small	0.83	0.78
MobileNetV3Large	0.85	**0.86**
NasNetMobile	0.84	**0.86**

Moreover, the model trained with augmented data does not only have low accuracy but their F1-score and recall also are imbalanced according to Figure 12, Figure 13, Figure 14 and Figure 15. As a result, the augmented data model does not classify well in all class as InceptionResNetV2 trained on augmented data has an F1-score on class df and the akiec is just above 0.3 and 0.4, separately, while InceptionResNetV2 trained on metadata and the new weight loss can classify well in a balanced way according to Figure 12 and Figure 13. However, only DenseNet201, InceptionResNetV2, and NasNetLarge whose depths are equal to or larger than 400 have balanced the F1-scores on class. The others still face the imbalanced term. Since this data set is not balanced, therefore using augmented data can make the model more biased to the class which has a larger sample. Although using the metadata still leads to model biased, it does contribute to the improvement of the performance of the model.

This problem is also true with the recall according to Figure 14 and Figure 15. DenseNet201 and InceptionResNetV2, trained with augmented data have expected recall values of 0.56 and 0.69, respectively, while the combination of DenseNet201, Metadata, and the new weight loss function achieve the expected value of recall of 0.82. Therefore, metadata do improve the model performance by reducing the amount of data needed for achieving higher results. On the other hand, the reason why the model becomes much more balanced is the weighted loss function. Weight loss function has the ability to solve the imbalanced class samples by adding a weight related to the number of samples in each class. DenseNet201 and InceptionResNetV2 trained with the new weighted loss function have recall in akiec of 0.85 and 0.82, respectively, as opposed to their training in akiec without weighted loss function: 0.65 and 0.37.

Another interesting point found during the experiment is that MobileNetV2, MobileNetV3, and NasNetMobile have a small number of parameters and depth, but they have relatively good performance. MobileV3large, MobileV3Small, NasNetLarge and NasNetMobile outperform others on classifying class df with the recall of 0.92, 1, 0.92 and 0.92, respectively, according to the Table A5 in Appendix C. It is obvious that MobileNetV3Large and NasNetMobile are the two best performance models. Nevertheless, MobileNetV3Large has fewer number of parameters and depth than NasNetMobile.

Table 6 shows that the MobileNetV3Large, although the number of parameters is much smaller than that of DenseNet201. InceptionResNetV2, achieves an accuracy nearly to the others. In detail, MobileNetV3Large whose number of parameters has 5.5 million parameters, which is four and ten times less than DenseNet201 and InceptionResNetV2, respectively. The depth of MobileNetV3Large, on the other hand, is four times less than DenseNet201, InceptionResNetV2 which are 118 hidden layers as opposed to the 402 and 449 values of DenseNet201 and InceptionResNetV2, separately. Although, MobileNetV3Larege only achieves an accuracy of 0.86, the time needed for prediction is 10 and 30 times less than the other opponents. Since MobileNetV3Large needs a harder process of parameter hyper-tuning to achieve a better result, this is also the future target of this research.

Table 7 shows the AUC of the three models—InceptionResNetV2, Densenet201, and ResNet50—which are trained with only augmented data or metadata. It is transparent that the InceptionResNetV2 and DenseNet201 have higher AUC trained with metadata: both 0.99 as opposed to 0.972 and 0.93, respectively. ResNet50 trained with augmented data, on the other hand, has a higher AUC of 0.95 as compared to 0.93 of ResNet50 trained with metadata. Overall, InceptionResNetV2 trained with metadata reaches the peak with an AUC of 0.974. The InceptionResNetV2 trained with metadata is also compared with the others to find out the best models trained. According to Figure 10, Figure 11 and Figure 16 the InceptionResNetV2 still hit the peak AUC of 0.99. In contrast, ResNet152 otherwise is the worst model with the AUC of 0.87. Other models, on the other hand, have the approximately the same AUC.

In addition to the comparison between the original weight loss calculated by the sample percentage of each class model and the new weight loss-based model, it is also conducted on the three best-performing models including InceptionResNetV2, DenseNet201, and MobileNetV3. After the experiment, it is found out that the new weight loss function does not only contribute to the model to overcome the data imbalance problem but it also makes the accuracy increase. The performance of models is described in Table 8.

According to Table 8, the InceptionResNetV2 is found to be the best model trained. Furthermore, the InceptionResNetV2 is compared with the other state of the art researched models. According to Table 9, there are six researchers that use the same data set: HAM10000 but they have different approaches. These models used in that research are also SOTA models sorted in ascending order. The table shows that the accuracy of the combination of InceptionResNetV2 with Soft-Attention, metadata, and weight loss in this research is less than that of InceptionResNetV2 with Soft-Attention and augmented data: 0.90 compared to 0.93 respectively. However, since Soumyyak et al. uses data augmentation for all class of an imbalanced data set, the F1-score and recall are much lower. This is because the model in that research can only classify well on NV and VASC classes, which have the highest number of samples. On the other hand, the InceptionResNetV2 in this research also outperforms the other models according to five indicators: accuracy, precision, F1-score, recall, and AUC.

However, there are still some drawbacks of the model: the InceptionResNetV2 cannot well classify the melanoma and the nevus. According to Figure 17 the model sometime classifies the black nevus as the melanoma because of the same color between them. However, this problem is not true for the hard black or big melanoma or the red black nevus. Some future approaches that can be proposed would be to change the type of color to the other to fix the same color problem.

## 4. Conclusions

In this work, we proposed a model formed by a combination of one backbone model and Soft-Attention. Moreover, the model takes two inputs, including image data and metadata. A new weight loss function is applied to figure out the data imbalance problem. Finally, the combination of InceptionResNetV2, Soft-Attention, and metadata is the best model with an accuracy of 0.9. Although the accuracy and the precision of the model are not the highest, the F1-score, recall, and AUC of 0.86, 0.81, and 0.975, respectively are the highest and the most balanced indicators. Therefore, InceptionResnetV2 can classify well in all classes including low-samples classes. Otherwise, during the experiment, the combination of MobileNetV3, Soft-Attention, and metadata achieves an accuracy of 0.86 that is nearly the same as InceptionResNetV2, although with fewer number parameters and depth. Therefore the infer time is much less than that of InceptionResNetV2. This result opens the door to constructing a great performance model that can be applied to mobile and IoT devices. As a result, the proposed method and others still face the problem of badly distinguishing between melanoma and black nevus because in some cases, the melanoma and the nevus image have the same lesion size and color.

## Figures and Tables

**Figure 1 sensors-22-07530-f001:**
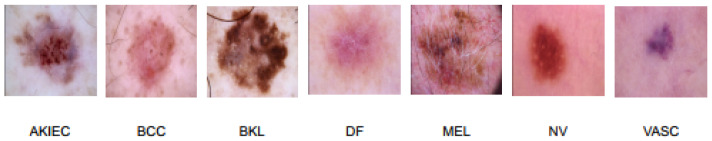
Example image of each class.

**Figure 2 sensors-22-07530-f002:**
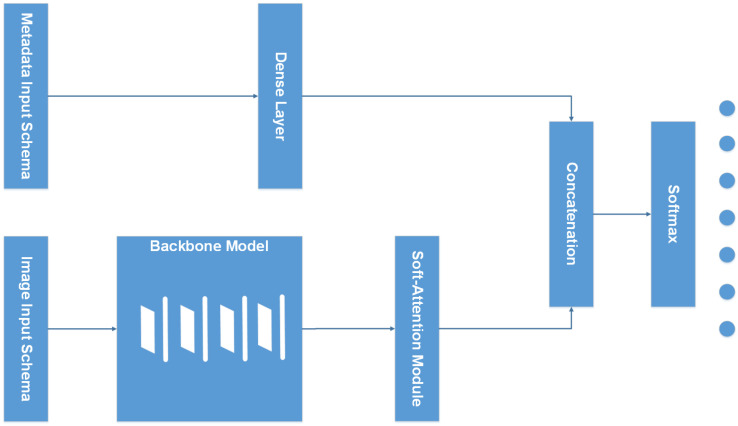
Overall model architecture.

**Figure 3 sensors-22-07530-f003:**
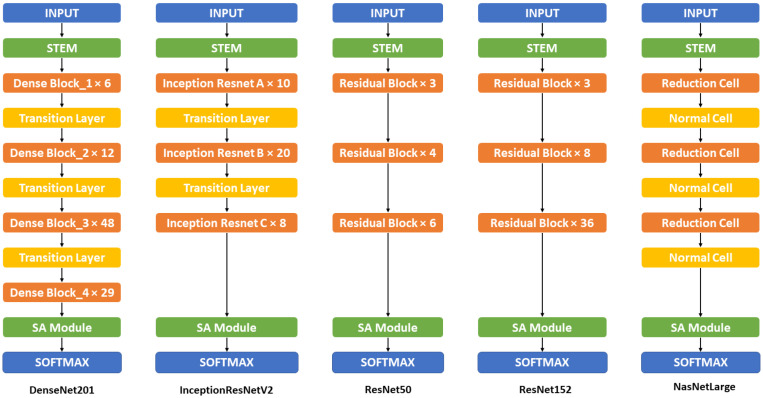
Proposed backbone model architecture. This figure show the overall structure of the backbone model (non mobile-based model) including DenseNet201, InceptionResNetV2, ResNet50, ResNet152, and NasNetLarge with Soft-Attention. The detailed structure and information can be found in the Table A1 in Appendix A.

**Figure 4 sensors-22-07530-f004:**
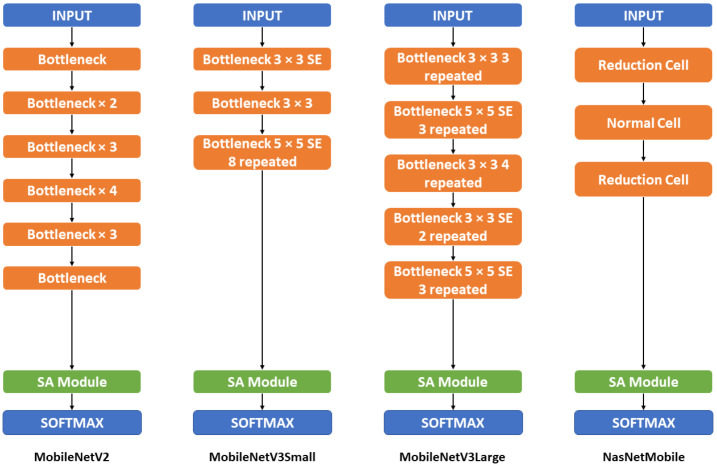
Mobile-based backbone model architecture. This figure shows the overall structure of the mobile-based backbone model including MobileNetV2, MobileNetV3Small, MobileNetV3Large, and NasNetMobile. The detailed structure and information can be found in the Table A2 in Appendix B.

**Figure 5 sensors-22-07530-f005:**
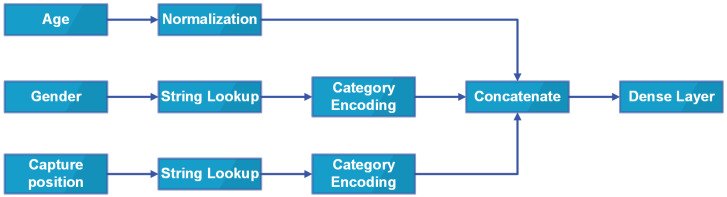
Input schema.

**Figure 6 sensors-22-07530-f006:**
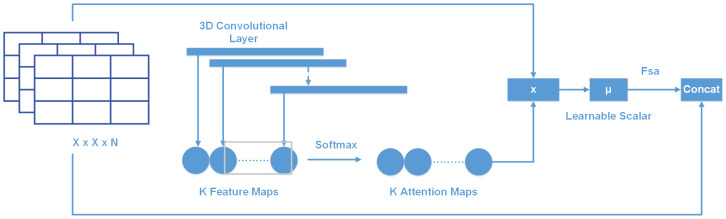
Soft-Attention layer.

**Figure 7 sensors-22-07530-f007:**
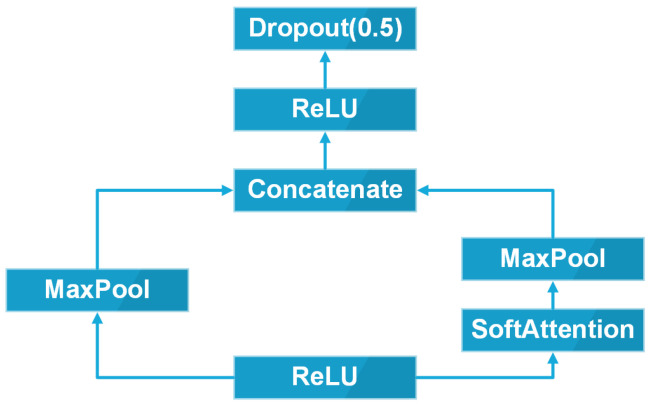
Soft-Attention module.

**Figure 8 sensors-22-07530-f008:**
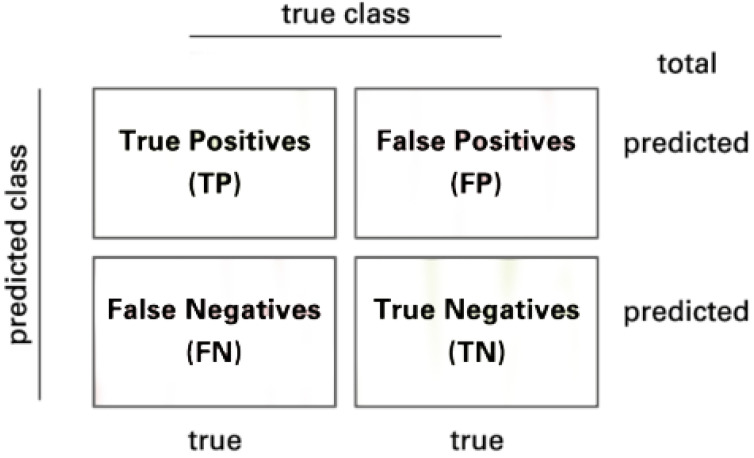
Confusion matrix.

**Figure 9 sensors-22-07530-f009:**
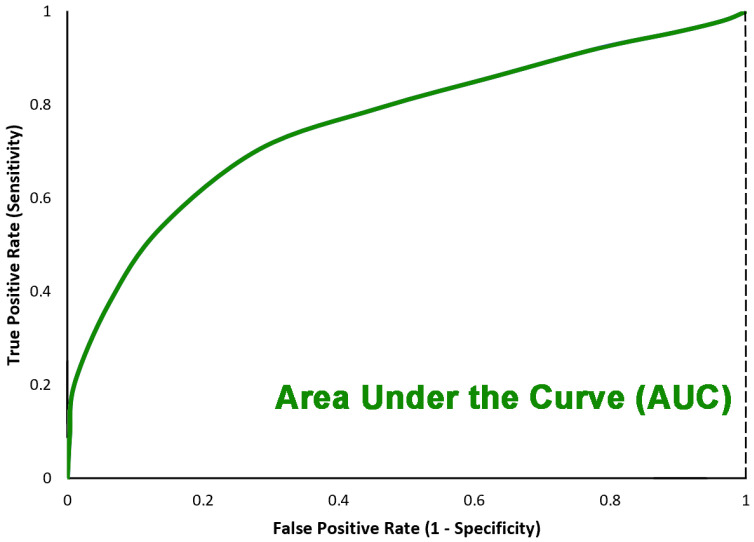
Area under the curve.

**Figure 10 sensors-22-07530-f010:**
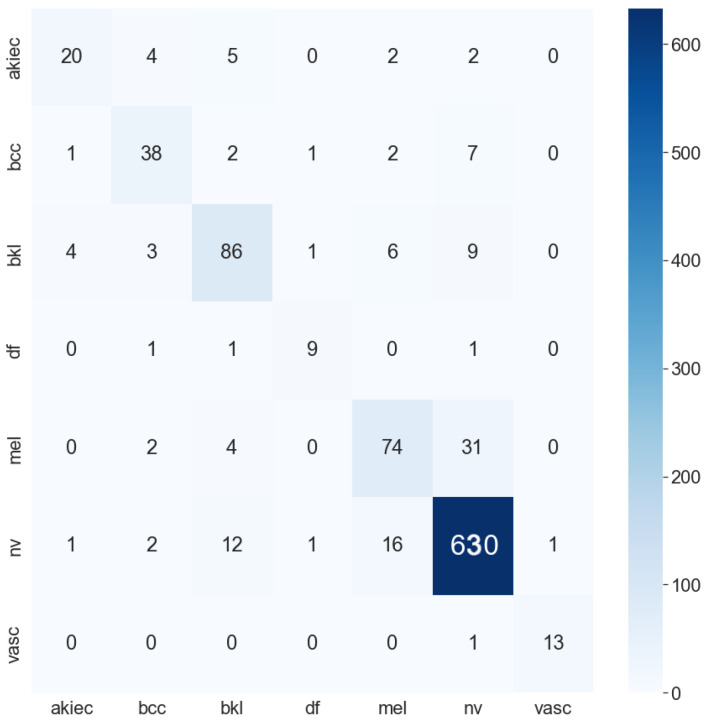
DenseNet201 confusion matrix.

**Figure 11 sensors-22-07530-f011:**
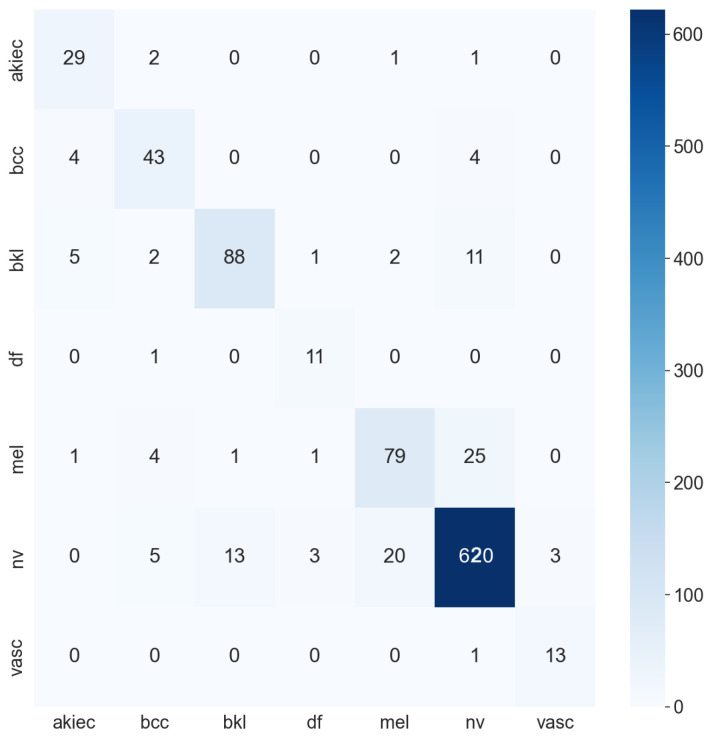
InceptionResNetV2 confusion matrix.

**Figure 12 sensors-22-07530-f012:**
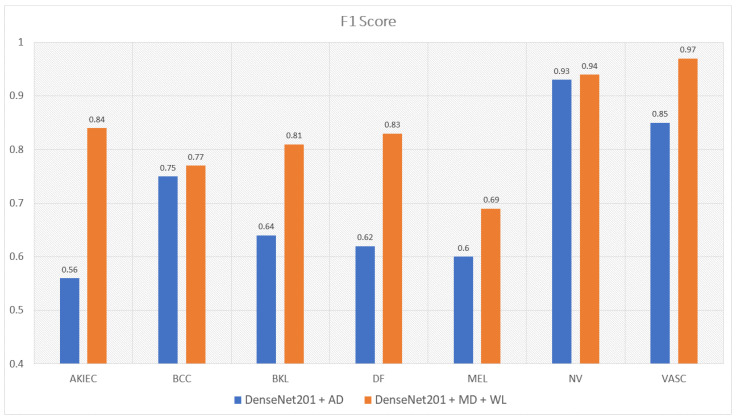
The comparison between F1-scores of DenseNet201 trained with augmented data and the one trained with metadata and weight loss.

**Figure 13 sensors-22-07530-f013:**
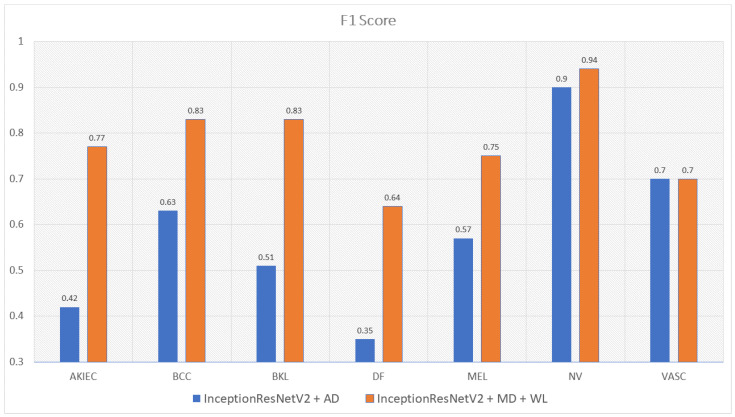
The comparison between F1-scores of InceptionResNetV2 trained with augmented data and the one trained with metadata and weight loss.

**Figure 14 sensors-22-07530-f014:**
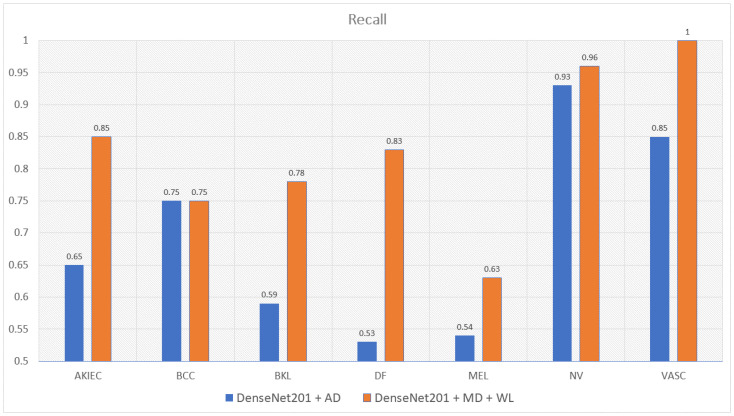
The comparison between recall of DenseNet201 trained with augmented data and the one trained with metadata and weight loss.

**Figure 15 sensors-22-07530-f015:**
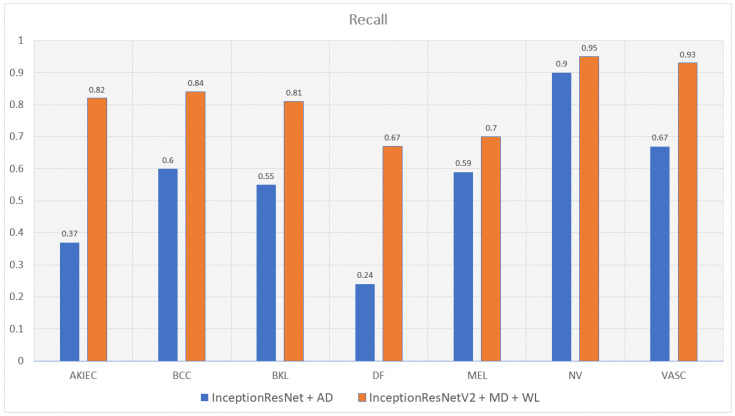
Comparison between recall of InceptionResNetV2 trained with augmented data and the one trained with metadata and weight loss.

**Figure 16 sensors-22-07530-f016:**
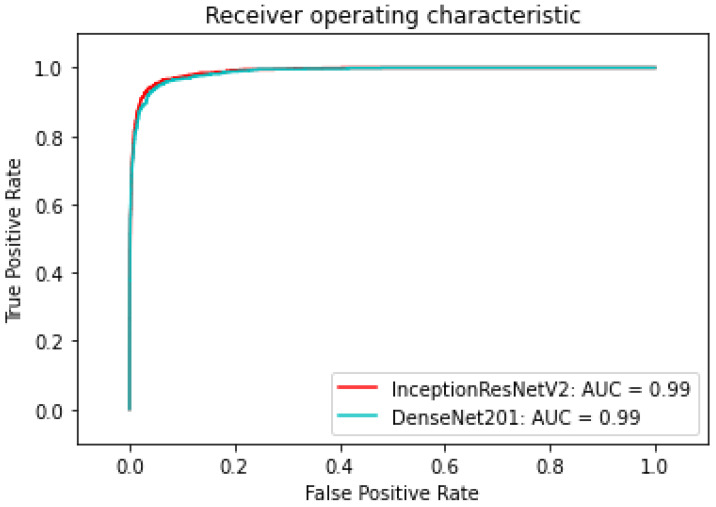
ROC of DenseNet201 and InceptionResNetV2.

**Figure 17 sensors-22-07530-f017:**
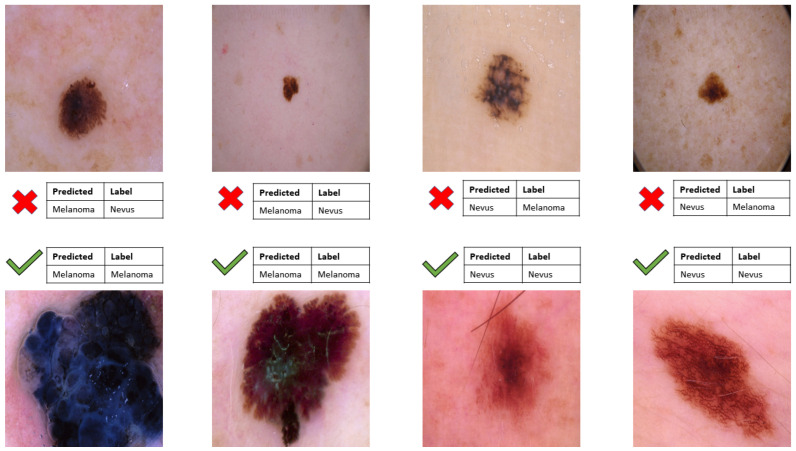
Model ability to classify melanoma and nevus.

**Table 1 sensors-22-07530-t001:** Summary of related works.

Work	Deep Learning	Machine Learning	Data Augmentation	Feature Extractor	Data Set	Result
[1]	Classify		x		HAM10000	0.93 (ACC)
[14]	Classify	Classify	x	x	HAM10000	0.9 (ACC)
[15]	Classify	Classify	x		HAM10000, *PH*^2^	
[16]	Classify		x		HAM10000	0.88 (ACC)
[17]	Classify		x		HAM10000	0.86 (ACC)
[18]	Classify		x	x	HAM10000, BCN-20000, MSK	0.85 (ACC)
[19]	Classify		x		HAM10000	0.85 (ACC)
[20]	Classify		x		HAM10000	0.92 (AUC)
[21]	Classify		x		HAM10000	0.92 (AUC)
[22]	Classify		x		HAM10000	0.74 (recall)
[23]		Classify	x	x	HAM10000	
[24]	Classify		x		HAM10000	0.92 (ACC)
[25]	Seg				HAM10000	0.99 (ACC)
[26]	Seg				HAM10000	0.97 (ACC)

**Table 2 sensors-22-07530-t002:** Data distribution in HAM10000.

Class	AKIEC	BCC	BKL	DF	MEL	NV	VASC	Total
No. Sample	327	514	1099	115	1113	6705	142	10,015

**Table 3 sensors-22-07530-t003:** Metadata example in the data set.

ID	Age	Gender	Local
ISIC-00001	15	Male	back
ISIC-00002	85	Female	elbow

**Table 4 sensors-22-07530-t004:** Size, parameters, and depth of the backbone model used in this paper.

Model	Size (MB)	No. Trainable Parameters	Depth
Resnet50	98	25,583,592	107
Resnet152	232	60,268,520	311
DenseNet201	80	20,013,928	402
InceptionResNetV2	215	55,813,192	449
MobileNetV2	14	3,504,872	105
MobileNetV3Small	Unknown	2,542,856	88
MobileNetV3Large	Unknown	5,483,032	118
NasnetMobile	23	5,289,978	308
NasnetLarge	343	88,753,150	533

**Table 6 sensors-22-07530-t006:** Comparison between MobileNetV3Large with DenseNet201 and InceptionResNetV2.

Model	MobileNetV3Large	DenseNet201	InceptionResnetV2
No. Trainable Parameters	**5,490,039**	17,382,935	47,599,671
Depth	**118**	402	449
Accuracy	0.86	0.89	0.90
Training Time (seconds/epoch)	116	1000	3500
Infer Time (seconds)	**0.13**	1.16	4.08

**Table 7 sensors-22-07530-t007:** AUCs of all models. AD stands for augmented data, this indicates that the model is trained with augmented data. MD stands for metadata, which indicates that the model is trained with metadata. Bold numbers highlight the highest performance.

Model	AUC (AD)	AUC (MD)
InceptionResNetV2	0.971	**0.99**
DenseNet201	0.93	**0.99**
ResNet50	0.95	0.93
ResNet152	0.97	0.87
NasNetLarge	0.74	0.96
MobileNetV2	0.95	**0.97**
MobileNetV3Small	0.67	0.96
MobileNetV3Large	0.96	**0.97**
NasNetMobile	0.96	**0.97**

**Table 8 sensors-22-07530-t008:** Loss-based model accuracy comparison.

Model	No Weight	Original Loss Accuracy	New Loss Accuracy
InceptionResNetV2	0.74	0.79	0.90
DenseNet201	0.81	0.84	0.89
MobileNetV3Large	0.79	0.80	0.86

**Table 9 sensors-22-07530-t009:** Comparative Analysis. Bold numbers highlight the highest performance.

Approach	Accuracy	Precision	F1-score	Recall	AUC
InceptionResNetV2 [1]	0.93	0.89	0.75	0.71	0.97
[14]	-	0.88	0.77	0.74	-
[16]	0.88	-	-	-	-
[17]	0.86	-	-	-	-
GradCam and Kernel SHAP [18]	0.88	-	-	-	-
Student and Teacher [19]	0.85	0.76	0.76	-	-
Proposed Method	0.9	0.86	**0.86**	**0.81**	**0.99**

## Data Availability

The code and the data analysis report can be found here: https://github.com/KhoiDOO/Skin-Disease-Detection-HAM100000.git (accessed on 29 August 2022).

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
