# Peer review of "Skin Lesion Classification on Imbalanced Data Using Deep Learning with Soft Attention"

_sensors, 2022, doi:10.3390/s22197530_

Round 1

Reviewer 1 Report

The paper proposes a deep learning based methodology for skin disease recognition from dermoscopy images. The results of experiments on a skin image database are presented. The paper needs to be revised according to the comments and suggestions provided below before it could be considered for publication.

Major Comments:

1.       The novelty and innovativeness of the paper must be outlined. The paper uses well-known deep learning models (DenseNet, InceptionNet, ResNet, NasNet, and MobileNet), which have been used several times before by different authors, including for the skin disease recognition tasks. The research paper should go beyond the application of known methods, which is a standard engineering task. The proposed methodology is traditional.

2.       The related works subsection is poorly organized and presented. The selection of works seems to be ad hoc. I suggest to add some structural organization (e.g., machine learning based, deep learning based methods) and discuss the state-of-the-art papers published in the previous 2-3 years, which better reflect the trends and achievements in this rapidly evolving research field. The authors are encouraged to discuss, for example, Malignant skin melanoma detection using image augmentation by oversampling in nonlinear lower-dimensional embedding manifold. Extraction of abnormal skin lesion from dermoscopy image using VGG-SegNet. Melanoma segmentation: A framework of improved DenseNet77 and UNET convolutional neural network. Finalize by discussing the limitations of existing methods as a motivation of your study.

3.       Provide specific values of image augmentation parameters for replicability.

4.       Explain how you set the hyperparameter values for training such as training epochs and batch size. Did you use any hyperparameter optimisation/finetuning?

5.       More experimental results should be added such as confusion matrices and ROC plots.

6.       Evaluate the computational complexity of the proposed methodology. Report on the total number of trainable parameters in the proposed model.

7.       Compare your results with the results of other studies using the same datasets.

8.       Add the discussion section and discuss the limitations of the proposed methodology.

9.       The conclusions section just summarizes all findings of this study.  What are the deeper implications of this study and its significance to the biomedical research field? Support your claims by the main numerical findings from this study.

Minor comments:

10.   Extend Table 1 by reporting more specific information about the discussed studies such as deep learning models used and accuracy (performance) achieved.

11.   The caption of Table 6 is confusing.

12.   Why there are missing values in Table 7?

Reviewer 2 Report

The current manuscript has some novelty in proposed contribution. The experimental results provide fair comparison. It needs revision in terms of technical details before acceptance. Consider following comments in the revised version;

1. The sentence “In this paper, a combination of SOTA model such as DenseNet, 8 InceptionNet, ResNet, NasNet, and MobileNet and Soft-Attention is proposed” in the abstract is not correct. You didn’t use combination of all them in a specific unique structure. Different combinations of these networks are used in your proposed approach. Discuss about it or correct this sentence.  

2. All of the equations should be numbered.

3. How do you propose the weights formula? (Section 2.2.5). Is there any related reference? How much is the size of output W in this equation?

4. As I know, the input size of the used CNNs such as mobileNetV2, mobileNetv3, ResNet, etc are not same. Do you resize all of the images to the same size to start process? Or do you run each CNN with different input size?

5. I think your proposed approach can be used widely in medical applications. For example, it can be used in DNA classification, etc. For example, I find a paper titled “DNA Repair Genes (APE1 and XRCC1) Polymorphisms–Cadmium interaction in Fuel Station Workers”, which has enough relation. Cite this paper and discuss about it as one the advantages of your proposed approach.

6. It is suggested to discuss about the runtime of your proposed method briefly ( Compare performance with other methods is not needed)

7. In scientific papers, usually, the title of the tables is written above them.

8. The pre-process of skin lesion recognition is skin detection process. For example, I find a paper titled “An innovative skin detection approach using color based image retrieval technique”, which has relation. Cit this paper and discuss about the necessary pre-process in this scope briefly.  

Round 2

Reviewer 1 Report

The authors have revised well. The manuscript can be accepted for publication.

Reviewer 2 Report

The authors have given satisfactory answers to most of the questions. Useful sections have been added to the text that enhance the understanding of the presented method. The article has enough innovation to be accepted.